# Predictors of Exercise Capacity in Dilated Cardiomyopathy with Focus on Pulmonary Venous Flow Recorded with Transesophageal Eco-Doppler

**DOI:** 10.3390/jcm10245954

**Published:** 2021-12-18

**Authors:** Carlo Caiati, Adriana Argentiero, Cinzia Forleo, Stefano Favale, Mario Erminio Lepera

**Affiliations:** Unit of Cardiovascular Diseases, Department of Emergency and Organ Transplantation, University of Bari, 70121 Bari, Italy; adrianaargentiero92@gmail.com (A.A.); cinzia.forleo@uniba.it (C.F.); stefano.favale@uniba.it (S.F.); marioerminio.lepera@uniba.it (M.E.L.)

**Keywords:** transesophageal echocardiography, cardiopulmonary exercise test, renal dysfunction, dilated cardiomyopathy, diastolic dysfunction, left ventricular filling pressure, obesity

## Abstract

The aim of this study was to clarify the relative contribution of elevated left ventricle (LV) filling pressure (FP) estimated by pulmonary venous (PV) and mitral flow, transesophageal Doppler recording (TEE), and other extracardiac factors like obesity and renal insufficiency (KI) to exercise capacity (ExC) evaluated by cardiopulmonary exercise testing (CPX) in patients with dilated cardiomyopathy (DCM). During the CPX test, 119 patients (pts) with DCM underwent both peak VO_2_ consumption and then TEE with color-guided pulsed-wave Doppler recording of PVF and transmitral flow. In 78 patients (65%), peak VO_2_ was normal or mildly reduced (>14 mL/kg/min) (group 1) while it was markedly reduced (≤14 mL/kg/min) in 41 (group 2). In univariate analysis, systolic fraction (S Fract), a predictor of elevated pre-a LV diastolic FP, appeared to be the best diastolic parameter predicting a significantly reduced peak VO_2_. Logistic regression analysis identified five parameters yielding a unique, statistically significant contribution in predicting reduced ExC: creatinine clearance < 52 mL/min (odds ratio (OR) = 7.4, *p* = 0.007); female gender (OR = 7.1, *p* = 0.004); BMI > 28 (OR = 5.8, *p* = 0.029), age > 62 years (OR = 5.5, *p* = 0.03), S Fract < 59% (OR = 4.9, *p* = 0.02). Conclusion: KI was the strongest predictor of reduced ExC. The other modifiable factors were obesity and severe LV diastolic dysfunction expressed by blunted systolic venous flow. Contrarily, LV ejection fraction was not predictive, confirming other previous studies. This has important clinical implications.

## 1. Introduction

Dilated cardiomyopathies (DCM) are an escalating problem in the modern medicine, aggravated by the still poor knowledge of the underlying etiology and associated with poor prognosis and disabling symptoms like dyspnea and reduced functional capacity [1]. Generally, a limitation in performing an aerobic effort is attributed to poor left ventricular contractile function. However, what specific resting cardiac abnormalities mediate the poor exercise performance is not clearly understood. Left ventricular (LV) diastolic dysfunction with elevated filling pressure probably has a prominent role, and is potentially more predictive than systolic dysfunction at rest as expressed by the left ventricular ejection fraction (LVEF) [2]. In particular, the physiology of restricted LV filling raises LV filling pressure to elevated values, transmitted back to the left atrium and the pulmonary capillary. This hemodynamic derangement increases the net filtration pressure, thus imbibing the pulmonary parenchyma and reducing pulmonary oxygen diffusion [3,4]. At the same time, and even more importantly, diastolic dysfunction with increased LV filling pressure may hamper the stroke volume increase during exercise since the increased filling flow that should mediate the increased stroke and cardiac output through activation of the Frank Starling law is opposed by high filling pressure [5]. Therefore, it appears crucial to properly estimate LV filling pressure in order to predict functional capacity.

Transesophageal Doppler echocardiography (TEE) is the best approach to predict LV diastolic filling pressure in patients with dilated cardiomyopathy (DCM) by optimally recording blood flow velocity in pulmonary veins [6]. A systolic fraction (S Fract) < 55% of the pulmonary venous flow (PVF) (the % of the integral of systolic flow to the integral of systolic plus diastolic flow) was found to be 91% sensitive and 87% specific in predicting the mean left atrial pressure > 15 mm Hg [7].

In addition, the atrial reversal duration, if longer than the transmitral A-wave, can reliably predict an increased LV telediastolic pressure; in the absence of an S Fract abnormality, this may predict an increase in LV pressure confined to the telediastole [8]. While a few studies have assessed the prognostic role of the pulmonary venous flow in patients with DCM [9,10,11], no study has demonstrated the potential of this functional parameter, optimally obtained by means of transesophageal Doppler echocardiography (TEE) [12,13], in predicting functional capacity expressed by peak VO_2_ [14].

Moreover, in patients with DCM, the relative role of renal insufficiency (KI) and obesity in reducing functional capacity is not properly known; KI impact on functional capacity has been mainly assessed in hemodialysis patients [15,16] and never evaluated along with solid LV diastolic functional parameters. Obesity and even nonterminal KI may increase the circulating volume, potentially impairing the diastolic function through a strain-dependent mechanism [17]; this may compound the effect of the intrinsic cardiac abnormality in limiting exercise endurance. We hypothesized that LV diastolic filling pressure, as assessed by PVF and mitral flow velocity Doppler recording through a transesophageal approach, along with other extracardiac factors (obesity and renal dysfunction), could be predictive variables of the functional capacity, objectively expressed by maximum oxygen consumption [18] in 119 patients with dilated cardiomyopathy in sinus rhythm.

## 2. Materials and Methods

The study was an observational prospective cohort analysis. Overall, 122 consecutive patients with newly diagnosed DCM in sinus rhythm and clinical stability were enrolled; their clinical and demographic characteristics are summarized in Table 1.

### 2.1. Study Criteria

We included dilated cardiomyopathies of all etiologies in this study. DCM was diagnosed on the basis of the following criteria: LV end-diastolic dimension > 60 mm, left fractional shortening < 25%, and an increased E-point septal separation obtained in the M-mode with cross-sectional echocardiographic guidance [19]. Ischemic cardiomyopathy was defined in the presence of documented previous myocardial infarction or a >50% luminal diameter stenosis on a major epicardial coronary artery at coronary angiography. Mitral regurgitation was visually estimated on a semiquantitative scale according to the maximum length and width of the abnormal jet relative to the left atrium as visualized in the four-chamber view [20,21]. Mitral regurgitation was considered severe in case the ratio of the jet area to the left atrium area was >50%, and at same time the jet caused a reversal of systolic flow in the pulmonary vein(s). Patients with severe regurgitation were excluded from the study.

All the patients were on the best-known therapy at that time.

The exclusion criteria were the presence of an acute cardiac or extracardiac illness, atrial fibrillation or any cardiac arrhythmia, severe valvulopathy, and, in particular, severe mitral valve regurgitation because this affects LV diastolic filling causing increased E-wave velocity.

All the patients were fully informed about the nature of the study and gave informed consent to take part.

### 2.2. Assessment

Fasting blood samples were taken to evaluate the routine hematological parameters and in particular kidney function and hemoglobin levels. Regarding renal function, glomerular filtration was assessed with the Cockcroft–Gault formula [22].

All the patients were subdivided by New York Heart Association (NYHA) Functional Classification (Table 1).

Cardiopulmonary exercise testing (CPX). All the patients who could undergo an effort test underwent an ergospirometric examination with a treadmill test using a modified Naughton protocol (T2000 treadmill, Marquette Electronics/Hellige, Milwaukee, WI, USA). We assessed the peak VO_2_ defined as oxygen consumption (mL/kg/min) at peak exercise calculated as the average VO_2_ value over the last 30 s of exercise. Measured peak VO_2_ was obtained with breath-by-breath analyses of expired gas (SensorMedics Co., Yorba Linda, CA, USA). A 12-lead electrocardiogram was continuously recorded and blood pressure was measured at every step of the exercise. The patients were encouraged to exercise until they reached at least the anaerobic threshold (AT), that is, the point when carbon dioxide production increases disproportionately in relation to oxygen consumption and the respiratory ratio is >1.0 [18].

### 2.3. Echocardiographic Evaluation

All the patients underwent a transthoracic echocardiographic examination with an HP Sonos 2500 ultrasound device (Hewlett Packard Co., Andover, MA, USA).

Standard two-dimensional echocardiography was performed to obtain the following parameters: left ventricular end-diastolic and systolic diameters (parasternal long axis view) and volumes (apical four-chamber view) with a modified Simpson rule, the inferior vena cava diameter in inspiration and expiration. Then, the left ventricular ejection fraction was calculated using volumes and the percentage of the inferior vena cava inspiratory collapse (diameter in expiration–diameter in inspiration/diameter in expiration).

The presence and magnitude of mitral valve regurgitation was quantitatively assessed by means of color Doppler, calculating the ratio of the mitral jet area to the left atrium area as visualized in the four-chamber view (patients with severe regurgitation were excluded from the study).

Transesophageal Doppler echocardiography. To evaluate LV filling pressure, pulmonary venous and mitral flow was recorded. To optimize PVF recording, a transesophageal approach was adopted [12,13]. Therefore, all the patients underwent transesophageal echocardiography with a 5/3.7 MHz omniplane probe (Hewlett Packard Co., Andover, MA, USA). The probe was introduced with the patient in left lateral decubitus after having been mildly sedated with 5–10 mg of intravenous diazepam and local pharyngeal anesthesia with a Xylocaine spray [23]. All the transesophageal Doppler echocardiographic examinations were performed by one researcher (C.C.).

During the TEE examination, first, the transmitral flow was recorded: under color guidance, the sample volume was placed first between the tips of the mitral leaflets, where the E/A ratio and E-wave deceleration time are determined best, and then at the annulus level, where the A duration is measured best [6]. Then, pulmonary venous flow (PVF) was recorded, placing the sample volume 1–2 cm within the orifice of the left upper pulmonary vein using a mid-esophageal view with an angle between 0° and 60° above the left atrial appendage [6]. In this way, we obtained a tri- or quadriphasic pattern consisting of the pulmonary first systolic wave (S1), when present, the second systolic wave (S2), the pulmonary venous early diastolic wave (D), and the atrial reversal flow wave (Ar).

All these measures were performed taking care to ensure the correct alignment of the ultrasound beam and the color Doppler diastolic signal flow, place the sample volume in the area of the laminar flow, and reduce the spectral broadening in order to avoid under- or overestimation of the velocities and time intervals [7].

The velocity curves were analyzed using the supplied commercial analysis system. The peak flow velocity and time–velocity integral during the forward systolic and diastolic flow, the deceleration time of the early diastolic flow, and the maximal velocity, velocity integral, and duration of flow reversal during atrial systole were measured as schematically illustrated in Figure 1.

The peak systolic and diastolic PVF wave ratio (S/D), mitral E- and A-wave ratio (E/A), mitral E deceleration time, and the time difference between the PVF atrial reversal Ar and the mitral A-wave duration (Ar-A) were calculated. In cases of double systolic waves, S2 was considered for the ratio. When the onset of the pulmonary Ar wave was difficult to determine (very rarely), we used the onset of the mitral A-wave because they are temporally coincident, as suggested by Appleton [14], or the difference in the A-wave duration that can be approximated by relating the end of the atrial flow phases to the QRS complex, whose onsets coincide [8]. In this way, Ar, the most critical point in the calculation of the difference in Ar-A duration, was measured well. Based on the combination of the mitral and pulmonary venous flow, in accordance with the guidelines [6,7,8,9,10,11,12,13,14,15,16,17,18,19,20,21,22,23,24], we identified three classes of progressively increased left ventricular filling pressure as reported and explained in Table 2: normal filling pressure, high isolated telediastolic left ventricular filling pressure, and high pressure before atrial contraction (or high mean atrial pressure).

All the examinations were recorded on videotapes, and the calculations were performed offline using the ultrasound equipment’s (HP Sonos 2500) built-in calculation software. We recorded Doppler signals during apnea at the end of expiration at a sweep speed of 50–100 mm/s and considered for calculations the mean of three beats.

### 2.4. Statistical Analysis

Continuous variables were expressed as the means ± SD, discrete variables—as absolute numbers and percentages. Independent samples *t*-test or Mann–Whitney U test were used as appropriate to compare continuous variables between group 1 (peak VO_2_ > 14 mg/kg/min) and group 2 (peak VO_2_ ≤ 14 mg/kg/min) patients. The effect size for the independent samples *t*-test was assessed by means of Cohen’s d effect size statistics: 0.2 = small effect, 0.5 = medium effect, and 0.8 = large effect. Comparisons between groups of discrete variables were performed by means of the χ^2^ or Fisher’s exact test if the expected cells count was <5. The S Fract best cutoff point was empirically estimated using the receiver operating characteristic (ROC). One-way ANOVA was performed to assess the effect of three different grades of diastolic dysfunction on peak VO_2_. Two-way ANOVA was performed to assess the interaction and the main effect of the LVEF and the inferior vena cava diastolic collapse (best dichotomized on the basis of the ROC) on creatinine clearance. The most clinically relevant variables with *p* < 0.05 in univariate analysis were then included as predictors (enter mode) in a logistic regression model. All the predictors were categorical in order to predict or explain our categorical dependent variable (peak VO_2_ categorized as > or ≤14 mL/kg/min). Statistical calculations were performed using IBM SPSS version 23 (IBM Corp.; Armonk, NY, USA).

## 3. Results

The study was initially carried out on 122 patients with dilated cardiomyopathy, all in sinus rhythm. The prediction of the left ventricular filling pressure best attained with the combination of transmitral and pulmonary venous flow Doppler recording identified 62 (51%) patients with normal left ventricular filling pressure, 20 (16%) with isolated increase in LV filling pressure after atrial contraction with presumably normal mean atrial pressure, and 40 (33%) with elevation of the left ventricular pressure before atrial contraction (Pre-a increase of pressure) (Table 2).

All the patients were scheduled for the cardiopulmonary exercise test with peak VO_2_ consumption assessment (Figure 2). Twenty-seven patients, however, were unable to perform the exercise due to either several comorbidities or acute heart failure (HF) decompensation. Of these, three were excluded from the analysis, while the remaining 24 (19%) were included on the basis of the accurate evaluation of functional capacity made through the NYHA class assessment. Thus, the final two subgroups consisted of 78 patients (64%) (first subgroup with peak VO_2_ > 14 mL/kg/min), including 19 patients who were unable to exercise but with a NYHA class ≥ II, and 41 patients (34%) (second subgroup with peak VO_2_ ≤ 14 mL/kg/min), including five patients unable to exercise but with a NYHA class III or IV (Table 3).

Univariate analysis. All the Doppler parameters investigating LV diastolic dysfunction (such as systolic/diastolic ratio of the PV flow velocity waveforms expressed either through the systolic fraction or S/D ratio, transmitral E/A ratio, transmitral E deceleration time) were significantly predictive of the peak VO_2_ consumption, with the exception of the isolated telediastolic restriction (Arv-A > 0) that was not predictive (Table 4).

When verifying which of these parameters had the largest difference between the groups with and without peak VO_2_ impairment, we found that S Fract, that indicates a more severe LV diastolic abnormality since it predicts an increased filling pressure in the passive filling phase before atrial contraction, had the largest effect size (Table 4). Thus, the systolic fraction of venous flow had the largest effect size and was also the most feasible parameter to assess LV diastolic restriction and function since transmitral parameters were not measurable in 15 patients due to the effect of tachycardia. An example of normal and abnormal S Fract is shown in Figure 3 and Figure 4.

Using ≤59% of S Fract as a cutoff, identified using ROC curve analysis (area under the curve = 0.69, 95% CI: 0.87–0.98; *p* < 0.0001), the sensitivity and specificity of predicting a reduced peak VO_2_ consumption were 61% and 72%, respectively (Figure 5).

To further analyze the prediction of the peak VO_2_ by Doppler indices of progressively rising LV filling pressure (no elevation = S Fract > 59% and A-Arev ≥ 0; telediastolic restriction = S Fract > 59% and A-Arev < 0; pre-A increase in pressure = S Fract ≤ 59%) in the group with the measured peak VO_2_ (88 patients), one way between-groups analysis of variance was performed. There was a statistically significant difference, with a medium–large effect size (η^2^ = 0.11) in the peak VO_2_ in three categories (F = 5.1, *p* = 0.008). Post hoc comparison showed that only a pre-A increase in pressure by S Fract identified a significantly lower peak VO_2_ than that predicted by just a telediastolic restriction and no restriction categories (Figure 6).

Other extracardiac parameters. Greater age, female gender, higher BMI, diabetes, and indices of renal dysfunction and anemia were also predictive of a worse cardiac functional capacity (Table 2).

We found that only a depressed forward systolic flow expressed by a severely impaired LV EF (<25%) and not-backward heart failure quantified by a severe reduction (<30%) of the vena cava inspiratory collapse (IVC-IC) explained a reduction in creatinine clearance. In fact, based on the two-way analysis of variance, the interaction effect between the LVEF and the IVC-IC was not statistically significant (F = 0.13, *p* = 0.71); there was a statistically significant main effect for the LVEF (F = 6.34, *p* = 0.01), with a considerable effect size (partial η^2^ = 0.05). Contrarily, the main effect for the IVC-IC did not reach statistical significance (F = 1.92, *p* = 0.17).

Multivariate logistic regression. Direct logistic regression was performed to assess the impact of the best predictors in univariate analysis of the likelihood of patients having a reduction of the peak VO_2_ (Table 5).

The model contained six independent variables (age, gender, left ventricular ejection fraction, creatinine clearance estimated with the Cockcroft-Gault formula, left ventricular diastolic function); two continuous variables (age and BMI) were collapsed in three categories with an approximately equal number of patients, while two other continuous variables (ejection fraction of the LV, clearance of creatinine) were dichotomized on the basis of the best cutoff attained with the ROC curves; lastly, the patients were subdivided into three groups for diastolic dysfunction on the basis of a progressive increase in the LV diastolic pressure estimate: group 1 (reference category), no sign of increase in the diastolic pressure; group 2, isolated telediastolic restriction; group 3, a pre-a pressure increase as estimated via the systolic fraction of PVF. The full model containing all the predictors was statistically significant (χ^2^ (6, *n* = 119) = 76.40, *p* < 0.001), indicating that the model was able to distinguish between the patients with the peak VO_2_ > or ≤14 mL/kg/min. The model taken overall explained between 44% (Cox and Snell R-squared) and 61% (Nagelkerke R-squared) of the total variance in the peak VO_2_ consumption. As shown in Table 5, all the included variables except EF made a unique, statistically significant contribution to the model. The strongest predictor of the peak VO_2_ was not a cardiac but a renal parameter, namely creatinine clearance, recording an odds ratio of 7.4. This indicated that the patients with an estimated creatinine clearance (Cockcroft-Gault equation) < 52.2 (previously obtained cutoff by means of ROC curve analysis) were 7.4 times more likely to have a peak VO_2_ ≤ 14 mL/kg/min than those with clearance > 52.2. Only the more severe diastolic dysfunction category resulted in a valid predictor of impaired peak VO_2_, identifying patients 4.9 times more likely than the reference category (no diastolic dysfunction) to have a reduced functional capacity, whereas an isolated telediastolic restriction did not. BMI in the higher category only (>28) was the third best predictor of impaired functional capacity. As expected, greater age and female gender were predictive of a reduced functional capacity. LV systolic function expressed by EF was not predictive of functional capacity in multivariate analysis.

## 4. Discussion

This study has clearly shown for the first time that elevated LV filling pressure as assessed by pulmonary flow velocity Doppler recording, rather than LV EF, can significantly explain a reduced functional capacity as objectively measured by CPX. In addition, other extracardiac factors have been shown to limit functional capacity: creatinine clearance < 52.2 mL/min was the best predictive variable, and then obesity, as indicated by BMI > 28. Age > 62 years and female gender were also predictive of a reduced functional capacity and completed the prediction model (Table 5).

Functional capacity is a major outcome in patients with DCM and retains a high clinical value [25]. Any effort to improve functional capacity is worthwhile in the DCM setting. It is not clear from the literature what factors contribute to hampering the functional capacity in patients with DCM since the studies so far have dealt with a few factors at a time or have assessed some parameters like LV diastolic function with improper techniques or procedures. In this study, we made an optimal assessment of the LV diastolic function by recording PVF velocity during TEE in order to understand the definite role of this parameter in predicting the functional capacity in patients with DCM. In addition, other extracardiac factors like renal function and obesity proved to be independent predictors of functional capacity. We believe there is a strong pathophysiologic connection between these independent predictors in reducing functional capacity owing to the expansion of blood volume mediated by obesity and RI, interacting with a restricted physiology of LV filling. The expansion of blood volume (by RI and obesity) induces more LV filling, thus abnormally increasing the diastolic pressure (strain-dependent diastolic dysfunction) since the LV diastole operates over a steep pressure–volume curve [17]. This situation deteriorates even further during physical exercise that triggers more venous return, causing a further increase in the LV filling pressure. This high filling pressure first creates more backward congestion, but more importantly, it limits the filling itself, thus preventing an adequate increase in cardiac output by the Frank Starling law, thereby hampering functional capacity. Contrarily, LV EF at rest is not predictive of the functional capacity, confirming other studies [3,4,5,6,7,8,9,10,11,12,13,14,15,16,17,18,19,20,21,22,23,24,25,26,27]. This is because notwithstanding the fact that LV systolic and diastolic function are inextricably interconnected, the contractile reserve (if any) cannot be used owing to the opposing high filling pressure [28].

### 4.1. LV Diastolic Filling Pressure Estimated via PVF

We tested, for the first time, TEE recording of venous flow to predict functional capacity in patients with DCM. In particular, isolated telediastolic restriction was a poor predictor of a reduced functional capacity both in univariate and multivariate analysis; this is not totally unexpected since the mean atrial pressure remains fairly normal in this pattern of diastolic dysfunction [8]. Nonetheless, this parameter may have prognostic implications that very few studies have pinpointed [9].

On the other hand, we found that a systolic fraction less than 57% predicted a functional capacity below 14 mL/kg/min best. This value in general predicts a moderate range of increased pre-A pressure [8]. Therefore, functional capacity starts to be hampered even with a mild to moderate augmentation of pre-a LV diastolic pressure. A major effort should be devoted to keeping the pre-A pressure low, as underlined in the clinical implications.

Regarding the Doppler parameters that predict filling pressure, we found not only a better performance of PVF, but also the systolic fraction of the pulmonary venous flow appeared to work better than the transmitral flow velocity and the more simplified S/D ratio of the pulmonary venous flow. Regarding the first point, mitral waves measurement was less feasible since fusion waves ensued in 21 patients. Moreover, the E/A ratio is a poor predictor of both wedge pressure and changes in the mean left atrial pressure, and for this reason, we believe, also of functional capacity. This stems from the fact that prolonged relaxation can maintain an E/A ratio < 1 even in the presence of elevated pre-a filling pressure; a condition that can be clarified by an abnormal S Fract that is based, instead, on the entire cardiac cycle (systolic and diastolic) of PVF. Regarding the second point, we believe that S Fract works better than the simplified S/D ratio since it takes into account all the complexities of the pulmonary venous flow, being based on measurements of flow velocity integrals both in the systole and in the diastole (i.e., how far blood travels during the time period). Systolic flow, in fact, is very complex. A reduction of systolic flow can be related either to the reduction of the first wave (S1) owing to impaired atrial relaxation or to the second wave reduction (S2, major wave) that is basically related to both poor LV ventricular contraction with a consequently minimal descent of the mitral annulus and/or also to inadequate right ventricular contraction that has insufficient strength to push the pulmonary venous flow forward from the back. So PVF, better delineated by S Fract, also takes into account the contractile performance of the right ventricle that has been shown to independently predict the functional capacity [29]. Remarkably, the superiority of S Fract over the S/D ratio is supported by a recent prognostic study [11].

More studies are needed to analyze the deceleration time of mitral E- and pulmonary D-waves.

### 4.2. Renal Insufficiency

Creatinine clearance < 52.2 mL/min was the best predictor of impaired functional capacity. We believe that the kidney-mediated major mechanism affecting functional capacity is volume expansion that finally brings about more filling pressure, limiting cardiac efficiency. Low cardiac output causes renal vasoconstriction, particularly of the efferent arterioles, via hormonal (angiotensin II locally produced) and catecholamine drive. On one hand, this tends to normalize the Bowman intracapsular pressure, but on the other, it causes a drop of hydrostatic pressure in the peritubular capillaries. Moreover, since the glomerular filtration rate declines, albeit less than the renal blood flow, the filtration fraction increases due to elevation of the oncotic pressure in the peritubular capillaries. These last two events cause a drop of transcapillary hydraulic pressure of the vasa recta that mediates more sodium reabsorption in the proximal tubule (where more than 60% of Na is normally reabsorbed) with a consequent blood volume expansion. In addition, the antidiuretic hormone level is constantly elevated since the reduced effective blood volume serves as a potent non-osmotic stimulus to enhance the release of the antidiuretic hormone (ADH) [30]. Finally, the intrarenal blood flow is redistributed from the cortical to the juxtamedullary nephrons that contain longer loops of Henle, with a greater sodium reabsorption potential [31].

The increased venous systemic pressure, or “back” pressure, can also reduce the glomerular filtration rate; congested veins can compress the tubules and increase the capsular pressure, thereby opposing filtration and finally reducing the glomerular filtration rate and sodium and water excretion [32]. However, our data did not support a role of venous congestion in reducing the renal function, confirming other studies reported in the literature [33,34]. We believe that “back” pressure may have a role only in cases of more severe, chronic systemic venous congestion that did not apply to our patients.

Apart from volume expansion, renal insufficiency can impair cardiac performance and functional capacity by means of a variety of other mechanisms: first of all, hypertension with an increased ventricular afterload, but also anemia that implies increased cardiac work and ionic alterations (metabolic acidosis, in particular) with potentially negative inotropic effects [35].

However, cardiac and renal diseases interact in a complex bidirectional and interdependent manner in both acute and chronic settings. In DCM patients, renal deterioration can progress independently of HF. In fact, the deterioration can be due to drugs and environmental toxicants [36,37] that are avidly reabsorbed in renal tubules. This process is strongly enhanced by hypoperfusion, so these chemicals reach very high concentrations in the tubules and interstitium [36,37,38]. Given the high oxidative stress exerted by these molecules on the renal tissue and the interstitium in particular, further renal damage might take place that can boost further renal dysfunction with an intrarenal mechanism (chronic interstitial nephritis) [39], especially if chronic diabetic nephropathy is present [40].

### 4.3. The Role of Obesity in Reducing Functional Capacity

Considerable evidence demonstrates the adverse effects of obesity on central and peripheral hemodynamics, as well as on cardiac structure and function. In particular, with exercise in class III obese patients, central blood volume increases by 20%, LV end-diastolic pressure increases by 50% (from an already elevated level of 21 mm Hg to 31 mm Hg), and LV dP/dt increases by 57% [41]. Fat-free (non-osseous) mass is thought to contribute to these alterations as augmentation of the total blood volume, and cardiac output cannot be accounted for by excess fat mass alone [41]. Thus, the expansion of blood volume induces strain-dependent diastolic dysfunction that brings about pulmonary congestion and pulmonary hypertension during exercise or even at rest [42]. In addition, most of these patients develop arterial hypertension that further worsens LV performance, thus inducing more pulmonary congestion. Other mechanisms of obesity verified in animal models that negatively affect the already compromised cardiac function (both diastolic and systolic) in DCM are related mainly to hormone derangements. They include lipotoxicity and lipoapoptosis, insulin resistance with hyperinsulinemia, leptin resistance and hyperleptinemia, reduced adiponectin levels, activation of the sympathetic nervous system, and activation of the RAAS [16]. Finally, especially in severe obesity, functional capacity can be further reduced by the presence of a restrictive lung disease with early onset of dyspnea during exercise [43].

### 4.4. Previous Studies

The major limitations of the previous reports are, firstly, that diastolic function has not been properly studied [5,6,7,8,9,10,11,12,13,14,15,16,17,18,19,20,21,22,23,24,25,26,27,28,29,30,31,32,33,34,35,36,37,38,39,40,41,42,43,44], and never by Doppler recording of PVF by means of a gold standard approach like TEE, and secondly, that no study has attempted to see the entire picture (cardiac and also extracardiac factors), instead focusing on specific abnormalities such as alveolar capillary diffusing capacity [3], peripheral muscle dysfunction [45], and so on, while never putting these abnormalities in context with an appropriate evaluation of LV diastolic function. Moreover, renal dysfunction, apart from in dialysis patients, has never been analyzed in previous studies in correlation with functional capacity in DCM [15]. On the other hand, poor prediction of functional capacity by means of the LVEF at rest in this study confirms previously reported data [5,6,7,8,9,10,11,12,13,14,15,16,17,18,19,20,21,22,23,24,25,26,27,28,29,30,31,32,33,34,35,36,37,38,39,40,41,42,43,44,45,46].

### 4.5. Clinical Implications

Major practical suggestions arise from these data on how to improve symptoms and possibly even prognosis. First of all, LV diastolic function must be primarily addressed with echocardiography. In cases where LV diastolic restriction is present and functional capacity is impaired, therapeutic intervention to ameliorate the LV diastolic function is warranted. The former can reverse the consequences of diastolic dysfunction (e.g., venous congestion), and the second can eliminate or reduce the factors responsible for diastolic dysfunction (e.g., myocardial hypertrophy, fibrosis, and ischemia) [17]. While the latter point is a long-term goal, the former can be pursued not only by diuretics, but by relying upon a very important but very often neglected intervention, that is, to reduce the heart rate. In fact, even a minimal prolongation of the diastolic period improves emptying of the congested pulmonary capillary bed, thus reducing the wedge pressure [17] and at the same time improving the nourishment of the myocardium by increasing the subendocardial blood flow [47] and hence improving contraction. This is especially true in patients with marked cardiac enlargement since in accordance with biophysical principles, the optimum heart frequency is an inverse function of its size [48]. Therefore, betablockers and, if necessary, ivabradine are very beneficial [49,50]. The reduction of weight in obese individuals is another natural, nonpharmacological strategy to address strain-dependent diastolic dysfunction since weight reduction reduces HR by reducing the sympathetic drive that is elevated in obese people and also reduces blood volume expansion [17]. Therefore, patients with cardiorenal syndrome, apart from restricting Na [51], that is the mainstay of treatment to avoid volume expansion [52], should have a reduced exposure to drugs and toxicants [37]. Therefore, chronic administration of drugs should, if possible, be kept to a minimum, and environmental chemicals [36] and toxicants in food [37,38,39,40,41,42,43,44,45,46,47,48,49,50,51,52,53] should be avoided. In particular, all processed food should be avoided since it is loaded with Na, and in addition, drugs proven to have a potential renal toxicity like statins [54], aspirin [55], and proton pump inhibitors [56] should be suspended or at least kept to a minimum dosage. If their suspension is not warranted, renal function must be very closely monitored.

### 4.6. Study Limitations

Several limitations are present in this study.

Even though the systolic fraction of PVF velocity is a very robust parameter in predicting LV restriction and now in predicting functional capacity, the diastolic deceleration time of the pulmonary D-wave, shown to have a very high prediction power for the pulmonary wedge pressure, was not assessed [57]. Although this parameter can be useful in atrial fibrillation [6], further studies should address this diastolic parameter of venous flow in predicting the functional capacity in patients with DCM.

We did not measure systematically the E/e’ ratio. This is a pretty robust parameter [58], but it has many pitfalls and should be replaced with other echocardiographic and even invasive measurements under a common clinical scenario, as recently suggested [59]. In particular, in DCM, it does not predict an isolated telediastolic increase in pressure like Arev-A does, a common finding in our study group (Table 2); in fact, E/e’ is an index of the mean pulmonary wedge pressure [24]. It does not work properly in patients with heart failure [60]; furthermore, it is imprecise in the left bundle branch block (LBBB), a rather common finding in our study group (the two examples, Figure 3 and Figure 4, in this paper had a LBBB) and in mitral annulus calcification that was also common in our patients. We think that pulmonary venous flow, if properly recorded, as it happens, especially with the use of contrast [61], in combination with transmitral flow is a better method than E/e’ in order to predict the LV filling pressure and functional capacity in DCM patients.

TEE is a semi-invasive approach that cannot be routinely used in DCM patients; therefore, this study should be considered a benchmark study for assessing the potential of the pulmonary venous flow in predicting functional capacity in DCM. Fortunately, a transthoracic approach has now shown feasibility of almost 100% in terms of recording the S- and D-waves and slightly less (90%) of the reversal wave, with a very tight correspondence with TEE recording [61]. The use of ultrasound contrast in very difficult chests can further improve the feasibility of recording PVF velocity [9,10,11,12,13,14,15,16,17,18,19,20,21,22,23,24,25,26,27,28,29,30,31,32,33,34,35,36,37,38,39,40,41,42,43,44,45,46,47,48,49,50,51,52,53,54,55,56,57,58,59,60,61,62]. Therefore, TTE can replace TEE in clinical practice in Doppler recording of PVF.

This study did not specifically address the prognostic impact of these parameters. However, S Fract and creatinine clearance have also been shown to have a major independent prognostic implication in a 13-year follow-up, as previously reported [63]. Further prognostic studies are needed in this regard.

The study explained at most 61% (Nagelkerke R-squared) of the total variance in the peak VO_2_ consumption. Therefore, other predictors are possibly missing. We believe that a reduced alveolar–capillary membrane diffusing capacity [3,4,5,6,7,8,9,10,11,12,13,14,15,16,17,18,19,20,21,22,23,24,25,26,27,28,29,30,31,32,33,34,35,36,37,38,39,40,41,42,43,44,45,46,47,48,49,50,51,52,53,54,55,56,57,58,59,60,61,62,63,64] and altered skeletal muscle response to exercise [65] should also be considered as independent predictors of functional capacity.

The model needs to be prospectively applied to demonstrate model validation [66].

## 5. Conclusions

A blunted systolic PVF, indicating a high pre-a LV filling pressure rather than a depressed LV EF has an independent role in predicting impaired functional capacity in patients with DCM. Cardiorenal syndrome and obesity in patients with DCM have a major impact on reducing functional capacity. This assessment has important clinical implications.

## Figures and Tables

**Figure 1 jcm-10-05954-f001:**
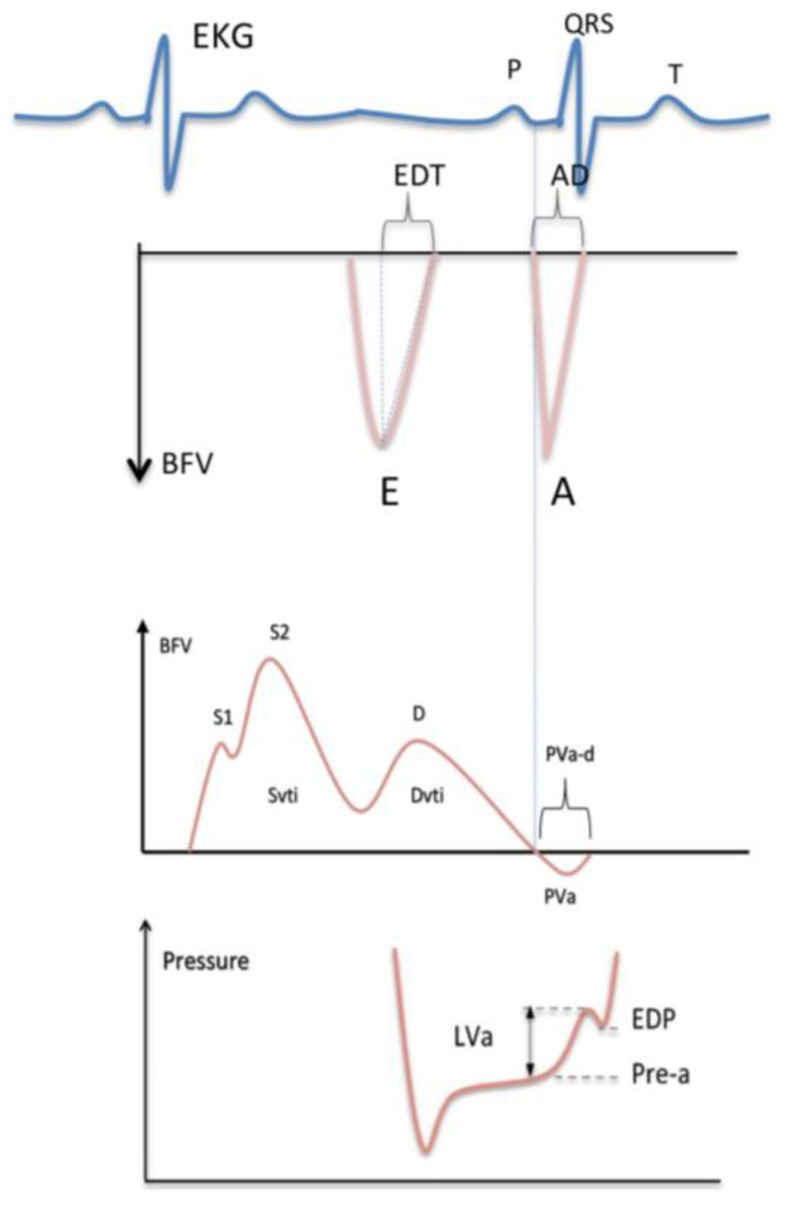
Measurement of the mitral flow velocity (**top**) and the pulmonary venous flow velocity (**middle**) with the left ventricular diastolic pressure curve (**bottom**). EKG = electrocardiogram; QRS = EKG waves related to ventricular depolarization; BFV = blood flow velocity; E = peak velocity of the early mitral flow; EDT = deceleration time of the early mitral flow; A = peak velocity of the late mitral flow; AD = duration of the late mitral flow; S = peak systolic velocity of the pulmonary venous flow; Svti = systolic velocity integral of the pulmonary venous flow; D = peak diastolic velocity of the pulmonary venous flow; Dvti = diastolic velocity integral of the pulmonary venous flow; PVa = peak velocity of the reverse flow at atrial contraction; PVa-d = duration of the reverse flow; LVa = increase in the ventricular pressure due to atrial systole; Pre-a = pressure before atrial contraction; EDP = left ventricular end-diastolic pressure. The vertical dashed lines indicate the simultaneous start of the antegrade flow through the mitral valve and the retrograde flow into the pulmonary vein at atrial contraction; the mitral waves (**top**) are negative since transmitral flow is away from the transducer from a transesophageal approach.

**Figure 2 jcm-10-05954-f002:**
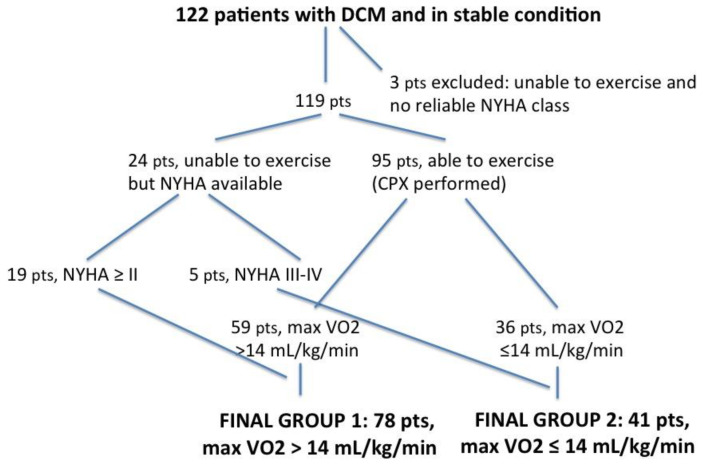
Flowchart explaining the enrolment criteria of our study group. DCM = dilated cardiomyopathy; pts = patients; NYHA = New York Heart Association functional classification; CPX = cardiopulmonary exercise testing.

**Figure 3 jcm-10-05954-f003:**
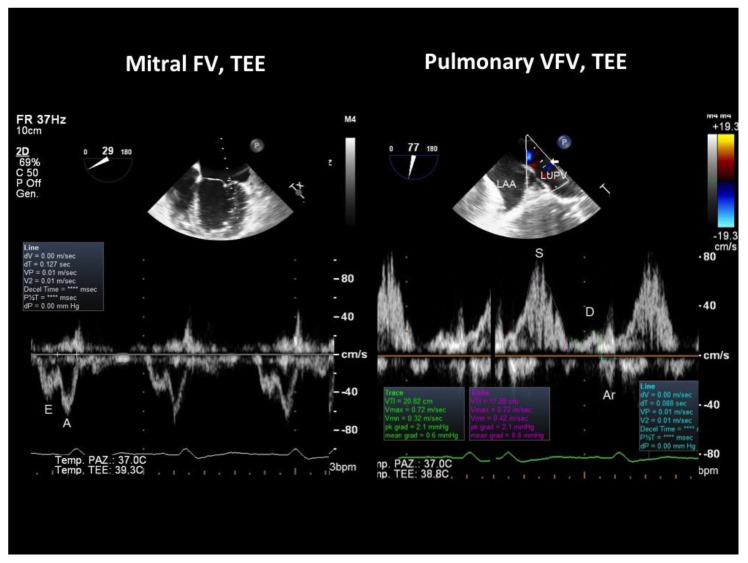
TEE recording of the mitral and pulmonary venous flow velocity in a patient with DCM showing a fairly normal peak VO_2_ (21 mL/kg/min). On the left, mitral flow velocity is displayed (bottom), and at the top, there is a 2D image showing the location of the sample volume in the LV (at the mitral annulus); the E/A ratio of the mitral waves was 0.7. On the right, the pulmonary venous flow is shown (bottom), and at the top, there is a 2D image showing the sample volume location 1 cm beyond the orifice of the left upper pulmonary vein with the left atrium, as indicated by the arrow: the systolic fraction of the venous flow was normal (82%); Arv-A was also normal (−39 ms), indicating a delayed relaxation of the LV with no increase in the filling pressure. TEE = transesophageal Doppler echocardiography; FV = flow velocity; VFV = venous flow velocity.

**Figure 4 jcm-10-05954-f004:**
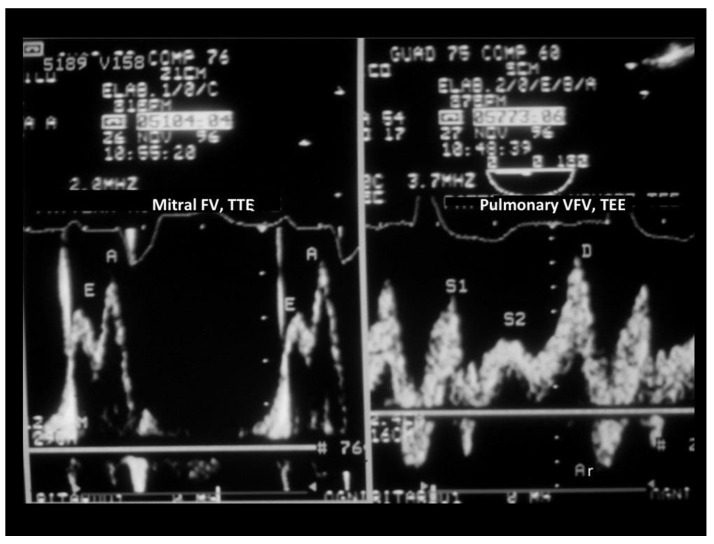
TEE recording of the mitral and pulmonary venous flow velocity in a patient with DCM and a reduced peak VO_2_ (12 mL/kg/min). On the left, mitral flow velocity is displayed (in this case, mitral flow was attained with a transthoracic approach); the E/A ratio of the mitral waves was 0.7, the same E/A ratio as in the patient in Figure 3. On the right, pulmonary venous flow is shown: the systolic fraction of the venous flow, differently from the case illustrated in Figure 3, was reduced (<50%), indicating this time a restriction of LV filling that predicts an increase in the Pre-a filling pressure. Arv-A was not computed as it is difficult to pinpoint the beginning of the mitral A-wave (precociously fused with the E-wave). There was a broadening of the pulmonary Doppler signal, so modal velocity was traced. TEE = transesophageal Doppler echocardiography; FV = flow velocity; VFV = venous flow velocity.

**Figure 5 jcm-10-05954-f005:**
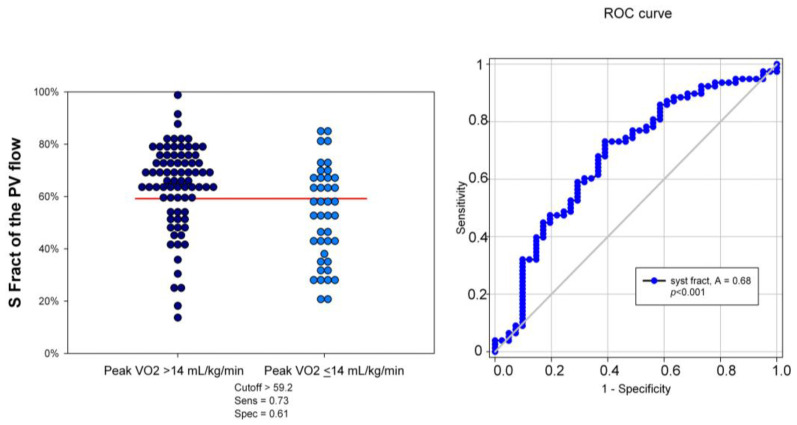
Dot histogram and ROC curve of S Fract in predicting the peak VO2 > and ≤14 mL/kg/min. The best S Fract cutoff was found at 59%, with an area under the curve of 68%, *p* < 0.001. Sens = sensitivity; Spec = specificity; S Fract = systolic fraction; PVF = pulmonary venous flow.

**Figure 6 jcm-10-05954-f006:**
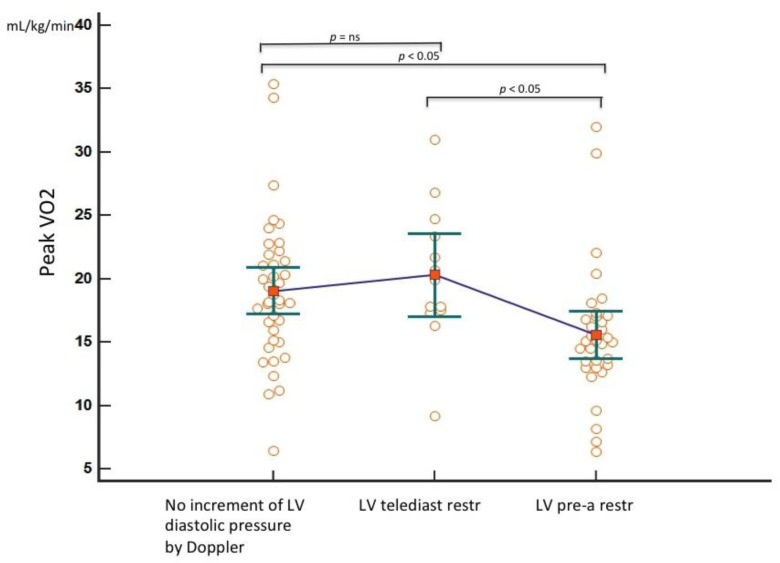
Means plot to compare the peak VO_2_ in three different categories of progressively increased LV filling pressure by TEE. Only the third group (pre-A increase in pressure) is significantly different from the first (no increase in LV filling pressure) and the second (isolated LV telediastolic restriction). For each group, the individual peak VO_2_ values with the mean and the standard error of the mean are reported. LV = left ventricle; telediast restr = telediastolic restriction (isolated increased telediastolic filling pressure); pre-a restr = increased pre-a pressure; ns = not significant.

**Table 1 jcm-10-05954-t001:** Demographics, clinical, echocardiographic, and CPX findings.

Study Group Findings		
Demographics		
Age (years)		59 ± 12
Sex	Male, *n* (%)	92 (75.4)
	Female, *n* (%)	30 (24.6%)
BMI		26.61 ± 3.74
Previous diseases		
Diabetes	Absent, *n* (%)	79 (64.8)
	Present, *n* (%)	43 (35.2)
DCM etiology	Nonischemic, *n* (%)	76 (62.3)
	Ischemic, *n* (%)	46 (37.7)
Renal insufficiency *	Absent, *n* (%)	87 (32)
	Present, *n* (%)	32 (27)
Blood tests		
Cholesterol (mg/dL)		192.47 ± 42.29
LDL (mg/dL)		119.13 ± 32.40
Na (meq/L)		139.73 ± 3.96
Cr (mg/dL)		1.25 ± 0.35
BUN (mg/dL)		0.51 ± 0.23
Hemoglobin (g/dL)		13.70 ± 1.70
Echocardiography		
LVEF (%)		29 ± 7
LV diastolic diameter (mm)		65 ± 7
LV systolic diameter (mm)		56 ± 8
CPX		
Peak VO_2_ (mg/kg/min)		18.17 ± 6.13
HR (b/min)		78.71 ± 13.00
NYHA class	1.00, *n* (%)	22 (18.8%)
	2.00, *n* (%)	37 (31.6%)
	3.00, *n* (%)	47 (40.2%)
	4.00, *n* (%)	11 (9.4%)

WEBER class	A, *n* (%)	29 (24%)
	B, *n* (%)	29 (24%)
	C, *n* (%)	30 (25%)
	D, *n* (%)	31 (26%)

Note: *n* = number of patients; * = no patients were on hemodialysis; BMI = body mass index; LDL = low-density lipoproteins; Na = plasmatic sodium; Cr = creatinine; BUN = blood urea nitrogen; LVEF = left ventricular ejection fraction; LV = left ventricle; CPX = cardiopulmonary exercise testing; HR = heart rate; NYHA = NYHA classification; WEBER = Weber classification based on peak VO_2_ and anaerobic threshold.

**Table 2 jcm-10-05954-t002:** Left ventricular filling pressure as predicted by means of transmitral and pulmonary venous flow Doppler recording in the initial study group of 122 patients.

LV Filling Pressure Categories by Doppler	
Normal LV FP, *n* (%)	62 (51%)
● E/A < 1 and Arev-A ≤ 0, *n* (%)	39 (32.8%)
● E/A ≥ 1 and < 2 and Arev-A ≤ 0 and SF > 55, *n* (%)	9 (7.6%)
● SF > 55% with E/A fusion, *n* (%)	14 (11.8%)
High LV TDFP, *n* (%)	20 (16%)
● E/A < 1 with Arev-A > 0, *n* (%)	17 (14%)
● E/A ≥ 1 and < 2 and Arev-A > 0 and SF > 55, *n* (%)	3 (2.5%)
High Pre-a FP, *n* (%)	40 (32.7%)
● SF < 55% with E/A > 2, *n* (%)	18 (15.1%)
● SF > 55% with E/A > 2, *n* (%)	2 (1.7%)
● SF < 55% with E/A fusion, *n* (%)	7 (5.9%)
● E/A ≥ 1 and <2 and Arev-A > 0 and SF < 55, *n* (%)	13 (10.1%)

LV = left ventricle; FP = filling pressure; TDFP = isolated telediastolic elevation of the filling pressure; Pre-a FP = filling pressure before atrial contraction; Arev-a = difference between duration of the pulmonary reversal A-wave and duration of the transmitral A-wave; E/A = ratio of the transmitral E-wave to the A-wave; SF = systolic fraction of the pulmonary venous flow.

**Table 3 jcm-10-05954-t003:** Variables predicting peak VO_2_ (univariate analysis).

Variables		Group 1	Group 2		
		VO_2_ Max > 14 mL/kg/min (*n* = 78)	VO_2_ Max ≤ 14 mL/kg/min (*n* = 41)	*p*	OR (95% CI)
Demographics and vital signs					
Age (years)		55 ± 10	66 ± 9	<0.001	−
Sex, *n (%)*	Male	66 (85.0)	23 (56.0)		1
	Female	12 (15.0)	18 (44.0)		4.3 (1.8–0)
BMI		26.09 ± 3.45	27.79 ± 4.13	0.03	−
SBP (mm Hg)		117.24 ± 11.89	117.88 ± 14.18	ns	−
DBP (mm Hg)		73.27 ± 7.42	74.73 ± 7.99	ns	−
HR (bpm)		79 ± 16.72	82.63 ± 14.45	ns	−
Max VO_2_		20.24 ± 5.34	11.31 ± 2.51	<0.001	
Risk factors					
DCM etiology, *n* (%)	Nonischemic	51 (65.0)	24 (58.0)	ns	1
	Ischemic	27 (35.0)	17 (42.0)	−	1.3 (0.6–2.9)
Diabetes, *n* (%)	−	59 (75)	19 (46%)	0.03	1
	+	19 (24%)	22 (54%)	−	3.6 (1.6–8.0)
Echocardiography					
EDD (mm)		65 ± 7	66 ± 7	ns	−
ESD (mm)		55 ± 8	57 ± 9	ns	−
EF (%)		30 ± 6	25 ± 6	<0.001	−
IVCc (%)		36.6 ± 13.2	23.3 ± 14.3	0.012	−
Mitral regurgitation (jet area, cm^2^)		18 ± 7	17 ± 8	ns	−
Mitral regurgitation (jet area/LA area, %)		34.6 ± 15	45.4 ± 18	0.004	
E/A fusion, *n (%)*	−	66 (85.0)	32 (78.0)	ns	1
	+	12 (15.0)	9 (22.0)		1.5(0.6–2.5)
E/A		1.14 ± 0.86	1.55 ± 0.91	0.035	−
EDT (s)		0.182 ± 0.08	0.148 ± 0.041	0.02	−
S Fract PV		63.81 ± 16.44	52.96 ± 18.12	0.001	−
S/D PV		1.32 ± 0.67	0.99 ± 0.65	<0.011	−
Ar PV-A (ms)		−8.64 ± 55.96	0.41 ± 67.62	ns	−
Blood tests					
Cr (mg/dL)		1.17 ± 0.25	1.40 ± 0.46	0.001	−
CrCl (mL/min)		69.83 ± 17.28	52.38 ± 18.00	<0.001	−
BUN (mg/dL)		0.45 ± 0.14	0.65 ± 0.31	<0.001	−
Hb (g/dL)		14.00 ± 1.32	12.97 ± 2.10	0.002	−
Na (mEq/L)		140.37 ± 3.58	138.60 ± 4.54	0.02	−
Drugs					
ACEi, *n* (%)	−	12 (15.0)	11 (27.0)	ns	1
	+	66 (85.0)	29 (71.0)	−	0.4 (0.1–1.2)
Diuretics, *n* (%)	−	17 (22.0)	1 (2.0)	0.009	1
	+	61 (78.0)	39 (95.0)	−	10 (1.4–84)
Digitalis, *n* (%)	−	34 (44.0)	8 (20.0)	0.016	1
	+	44 (56.0)	32 (78.0)	−	3.1 (1.3–7.6)
Nitrates, *n* (%)	−	52 (67.0)	24 (59.0)	ns	1
	+	26 (33.0)	16 (39.0)	−	1.3 (0.6–2.9)
BB, *n* (%)	−	58 (84)	34 (87)	ns	1
	+	11 (16)	5 (13)		0.8 (0.2–2.4)

OR = odds ratio; CI = confidence interval; ns = not statistically significant; VO_2_ max = maximal oxygen consumption; SBP = systolic blood pressure; DBP = diastolic blood pressure; HR = heart rate; DCM = dilated cardiomyopathy; EDD = left ventricular end-diastolic diameter; ESD = left ventricular end-systolic diameter; EF = ejection fraction; IVCc = collapsibility of the inferior cava vein; LA = left atrium; E/A = ratio of the peak values of the transmitral E- and A-waves; EDT = E-wave deceleration time; S Fract PV = systolic fraction of the pulmonary venous flow velocity; S/D PV = ratio of the peak pulmonary venous S- and D-waves; Ar PV-A = difference in duration between the reverse pulmonary venous and forward mitral A-waves; Cr = creatinine; CrCl = creatinine clearance according to the Cockcroft–Gault equation; BUN = blood urea nitrogen; Hb = hemoglobin; Na = plasmatic sodium; ACEi = angiotensin-converting enzyme inhibitor; BB = beta blockers; + = finding present; − = finding absent.

**Table 4 jcm-10-05954-t004:** Difference and effect size of the difference in the main Doppler parameters between the two groups, with the maximum VO_2_ >14 vs. ≤14 mL/kg/min.

Doppler Parameters	Sample Size	*t*-Test	Significance (Two-Tailed)	Mean Diff	95% CI of the	Cohen’s d
					Lower/Upper Difference	(Effect Size)
S Fract (%)	119	3.30	0.001	10.84	4.33/17.35	0.62
E DT (s)	98	2.85	0.005	0.034	0.0105/0.058	0.55
E/A ratio	99	2.13	0.035	−0.404	−0.780/−0.28	0.45
S/D ratio	119	2.58	0.011	0.329	0.076/0.582	0.50
A rev–A dur (ms)	98	0.700	0.486	−9.0	−34.68/16.60	0.14
A rev–A dur (ms)	98	0.700	0.486	−9.0	−34.68/16.60	0.14

S Fract = systolic fraction; E DT = deceleration time of the transmitral E-wave; E/A = ratio of the transmitral E- and A-waves; S/D ratio = ratio of systolic over diastolic pulmonary venous waves; A rev–A dur = duration of A-wave reversal in the pulmonary veins – duration of the transmitral A-wave; CI = confidence interval; Diff = difference; mean diff = mean difference between the group of the maximum VO_2_ > 14 mL versus the group with the maximum VO_2_ ≤ 14 mL.

**Table 5 jcm-10-05954-t005:** Logistic regression predicting the likelihood of a VO_2_ max ≤ 14 mL/kg/min.

Variables	B	Wald	df	*p*	Odds Ratio	95% CI for OR (Lower/Upper)	
Age ≤ 54 (reference category)		4.569	2	0.102			
Age > 54 and ≤62 years	0.71	0.822	1	0.365	2.051	0.434	9.690
Age > 62 years	1.71	4.558	1	0.033	5.531	1.151	26.593
Gender (reference category = male)	1.96	8.227	1	0.004	7.161	1.865	27.492
BMI ≤ 24 (reference category)		5.512	2	0.064			
BMI > 24 and ≤28	0.51	0.407	1	0.523	1.665	0.348	7.975
BMI > 28	1.77	4.776	1	0.029	5.881	1.201	28.806
LVEF (reference category > 25%)	0.83	1.775	1	1.183	2.296	0.676	7.793
Diastolic dysfunction (reference category, no diastolic dysfunction)		6.706	2	0.035			
Diastolic dysfunction (telediastolic restriction)	−0.18	0.030	1	0.862	0.829	0.099	6.906
Diastolic dysfunction (diastolic restriction)	1.59	5.234	1	0.022	4.934	1.257	19.366
CrCl by C.G. (reference category > 52.2 mL/m)	2.00	0.240	1	0.007	7.394	1.722	31.756
Constant	−4.68	16.540	1	0.000	0.009		

B = coefficient; df = degree of freedom; CI = confidence interval; BMI = body mass index; LVEF = left ventricular ejection fraction; diastolic restriction = pre-a increase in the left ventricular pressure; CrCl = creatinine clearance; C.G. = Cockroft–Gault equation; the p vales and odds ratios in bold refer to significant predictive variables.

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
