# Peer review of "Predictors of Exercise Capacity in Dilated Cardiomyopathy with Focus on Pulmonary Venous Flow Recorded with Transesophageal Eco-Doppler"

_jcm, 2021, doi:10.3390/jcm10245954_

Round 1

Reviewer 1 Report

Thank you for the opportunity to review the manuscript entitled: “Pulmonary venous flow in predicting functional capacity in dilated cardiomyopathy. A transesophageal echocardiographic doppler study” written by Caiati et al.

Predicting exercise intolerance in heart failure patients is a very important topic. The study was conducted properly. The topic is interesting but needs some clarifications and the manuscript should be improved.

Major comments:

  1. The information from the title is not supported by the results section.
  2. There is a lack of clinical implication of these findings.
  3. It should be clearly stated how diastolic dysfunction was assessed and grades of diastolic dysfunction should be presented according to the current guidelines.
  4. Why do authors use TEE to predicting left ventricular diastolic dysfunction and exercise capacity rather than TTE? LVEDP can be estimated from transthoracic echocardiography by assessing E/e’ ratio at rest and in some cases in exercise. Stress echocardiography could reveal elevated left ventricular feeling pressure not present at rest. Transesophageal echocardiography is an invasive procedure and the risk of complications restricts its wide use for diastolic dysfunction assessment. It should be clearly explained.
  5. Why authors did not measure pulmonary venous flow from TTE examinations. Was it technically possible?
  6. Why e’ from TDE was not recorded?
  7. Mitral regurgitation was assessed as jet area. Why not by PISA method which is recommended for quantitative assessment?
  8. Why patients were divided above and below 14 ml/kg/min peak VO2? How were distributed parameters according to Weber's classification?
  9. There is a lack of cardiopulmonary exercise test parameters: peak RER, VO2 at anaerobic threshold, VE/VCO2 slope, percent predicted peak VO2, Heart rate response, chronotropic index.

Minor comments:

  1. Exclusion criteria:

“Exclusion criteria were the presence of acute cardiac or extracardiac illness, atrial fibrillation or any cardiac arrhythmia, severe valvulopathies. (We also excluded severe mitral valve regurgitation because this affects LV diastolic filling, causing an increased E-

wave velocity).” Severe valvulopathies include severe MR. Criteria for severe mitral regurgitation should be described.

  1. English should be improved.

Author Response

ANSWERS.

Reviewer#1: The information from the title is not supported by the results section.

Response#1: Thanks for this comment. I agree that the title could not reflect the global message reported in the results section; so as suggested we changed the title but in keeping the message that this study focuses on pulmonary venous flow recorded by transesophageal Doppler echocardiography. (the new title in red page 1) :” PREDICTORS OF EXERCISE CAPACITY IN DILATED CARDIOMYOPATHY WITH FOCUS ON THE PULMONARY VENOUS FLOW RECORDED WITH TRANSESOPHAGEAL ECO-DOPPLER”

Reviewer#2: There is a lack of clinical implication of these findings.

Response#2: Thanks again for this comment. However I am sorry that we don’t understand this point of view of yours. We presented, in fact, a large list of very important clinical implications of our findings: the importance of keeping a low heart rate by beta blockers and eventually ivabradine and exercise in order to improve LV diastolic function, the reduction of weight, the importance to preserve and improve renal function essentially by reducing toxicants that are bad for the kidneys and by increasing hydration. For the first time we can suggest a very practical list of modifiable predictors of functional capacity and also some hints on how this predictors can be improved so leading to improving functional capacity that means the well being of the patients with DCM

Reviewer#3: It should be clearly stated how diastolic dysfunction was assessed and grades of diastolic dysfunction should be presented according to the current guidelines.

Response#3: We are grateful for this suggestion. As it has been stated in the guidelines, in patients with depressed LVEF like in DCM, LV filling pressure is what clinically is worthwhile to evaluate: quote:”the main reason for evaluating diastolic function in patients with reduced EFs is to estimate LV filling pressure”[1]. To this purpose as clearly expressed in the guidelines we consider the combination of two parameters the transmitral waves and the pulmonary venous flow. Even though in the guidelines in cases of difficulty in interpreting E/A ratio (E/A ratio >0.8 and <2) as first aiding parameters are suggested peak velocity of TR jet by CW Doppler, E/e’ ratio and left atrium (LA) maximum volume index, and only in case of unavailability of these parameters to recur to pulmonary venous flow, we used at the first hands the pulmonary venous flow for the high feasibility (100%) of this parameter with a transesophageal approach that was the approach adopted in our study. In fact in the guidelines the main reason to use the pulmonary venous flow as a second step approach is only due to the reduced feasibility of the venous flow Doppler recording especially in patients recovered in intensive care unit that is not the case in our high feasibility TEE study in stable patients. So we classified our patients on the basis of 3 criteria by combining mitral flow and pulmonary venous flow.

1) Normal filling pressure. It was either:

-an E/A <1 + Arev-A ≤ 0 or

-E/A ≥1 <2 + Arev-A ≤ 0 and SF >55 or

-SF >55% with E A fusion;

even though in the third case we are not totally sure about the presence of an isolated increase of the telediastolic pressure in some cases, we tended to consider that a minor problem in our study since to the purpose of prediction of exercise capacity, an isolated telediastolic restriction worked the same way as a normal filling pressure (Fig. 5).

2) Elevated isolated LV telediastolic pressure. It was either:

- E/A <1 with Arev-A >0

- E/A >=1 and <2 and Arev-A >0 and SF>55

3) Elevated pre-A left ventricular filling pressure (or mean left atrial pressure). It was either:

-E/A>2

- E/A >=1 and <2 and Arev-A >0 and SF<55

- SF <55% with E A fusion

We use as cutoff of Arev-A for telediastolic restriction a value > 0 rather than >30 msec. We chose this value in accordance with the landmark work of L. Hatle on this topic[2]. While Arev-A >0 was validated vs pressure Arev-A>30 msec value was mainly validated vs prognosis. In addition the high quality of our TEE method in recording the reversal A wave allows very accurate measurement of waves duration so even minimal variation can be spotted. We used also a systolic fraction of venous flow <55% to predict an elevation of pre-A LV filling pressure /mean left atrial pressure in accordance with the fact that such a cutoff was validated vs pressure in another landmark study that used the same TEE approach [3].

So in our population this 3 categories of predicted LV filling pressure were obtained as reported in the new Table 2 in the text:

Accordingly we have integrated the method (page 5, in red) : “Based on the combination of mitral and pulmonary venous flow in accordance with the guidelines[1,4] we identified 3 classes of progressively increased left ventricular filling pressure as reported and explained in Table 3: normal filling pressure, high isolated telediastolic left ventricular filling pressure and high pressure before atrial contraction (or high mean atrial pressure).

and the results section (page 6 in red) where we have also added the above reported table:” The prediction of left ventricular filling pressure best attained with the combination of transmitral and pulmonary venous flow Doppler recording identified 63 (53%) patients with normal left ventricular pressure, 18 (15.1%) with isolated increase of pressure after atrial contraction with presumably normal mean atrial pressure and 38 (32%) with elevation of left ventricular pressure before atrial contraction that coincides with an elevated mean left atrial pressure (Table 2).

Reviewer#4: Why do authors use TEE to predicting left ventricular diastolic dysfunction and exercise capacity rather than TTE? LVEDP can be estimated from transthoracic echocardiography by assessing E/e’ ratio at rest and in some cases in exercise. Stress echocardiography could reveal elevated left ventricular feeling pressure not present at rest. Transesophageal echocardiography is an invasive procedure and the risk of complications restricts its wide use for diastolic dysfunction assessment. It should be clearly explained.

Response#4:

Thanks for this observation. Since the aim of Doppler study was to predict LV filling pressure essentially by pulmonary venous flow Doppler recording that could have limited feasibility during TTE, and being a TEE approach a sort of gold standard for this kind of recording, we chose a TEE approach in order to shed a light on the definitive role of the pulmonary venous flow in predicting LV filling pressure. So this study is a kind of a benchmark study regarding the role of pulmonary venous flow in assessing LV filling pressure/mean left atrial pressure even though it was compared with functional capacity and not directly with pressure (wedge , atrial pressure etc.). However TTE now, thanks also to the advent of contrast, is really comparable with TEE in terms of feasibility of PVF recording as demonstrated some time ago. So we think that our data could be replicated with a TTE approach. In fact in good hands , with a good equipment or in more difficult cases using contrast enhancement, TTE works as TTE in Doppler recording of PVF.

So in in the manuscript as requested we have further explained this point in the limitations section of the discussion (page 16, in red): ‘TEE is a semi-invasive approach that cannot be routinely used in DCM patients; so this study has be considered a benchmark study for assessing the potential of pulmonary venous flow in predicting functional capacity in DCM. Fortunately, a transthoracic approach has now shown a feasibility of almost 100% in terms of recording the S and D waves and slightly less (90%) of the reversal wave with a very tight correspondence with a TEE recording [5]. The use of ultrasound contrast in very difficult chests can further improve the feasibility of recording PVF velocity [6,7]. So TTE can replace TEE in clinical practice in Doppler recording of PVF’.

Reviewer#5: Why authors did not measure pulmonary venous flow from TTE examinations. Was it technically possible?

Response#5: Thanks again for asking. Absolutely yes, TTE was a possible approach but with less accuracy and feasibility especially regarding Arev; in fact the ultrasound beam with a TTE approach is strongly attenuated especially in a big heart like in DCM when you insonify the pulmonary veins from the apex ; consequentially the atrial contraction movement causes wall thump strongly impairing Doppler recording of a weak flow signal like the reversal A wave, especially in its terminal part; in addition at the time of the study no contrast was available to increase the pulmonary venous flow signal.

Reviewer#6: Why e’ from TDE was not recorded?

Response#6: Thanks for this appropriate question; we did not systematically measured E/e’ first because our aim was to clarify the role of pulmonary venous flow with and without transmitral flow in predicting filling pressure and consequently functional capacity in DCM with an optimal Doppler recording attained from a transesophageal approach; secondly the E/e’ has not the potential to infer on the isolated telediastolic increase of pressure[4]. That prediction is only in the realm of pulmonary venous flow. Ar-A duration if > 0 can realibly predict an increase of telediastolic pressure[2]. This is a particular useful parameter since is the only age-independent of LA-A wave pressure increase and can separate patients with abnormal relaxation into those with normal filling pressure and those with elevated filling pressure but normal mean left atrial pressure (see the recently added table 2 in the revised manuscript). Not unexpectedly this parameter has got capability to predict outcomes.

Contrary E/e’ parameters has lots of limitations especially in patients with depressed left ventricular function; in particular it does not predict an isolated telediastolic increase of pressure like Arev-A does; in fact E/e’ is and index of mean wedge pressure that does not predict an isolated telediastolic pressure that in fact is associated with a normal mean left atrial pressure. It does not work properly in patients with hart failure ; also it is imprecise in LBBB, that was pretty common in our study group (the 2 examples in the paper have got a LBBB!) and in mitral annulus calcification that was common (even if not reported) in our patients; other conditions that can affect the reliability of this parameter are: mitral stenosis, prosthetic mitral valves, and surgical rings; it doesn’t work properly in pacemaker induced rhythm. When this ratio is >6 and <15 (most of the time), it needs to be integrated with other parameters that are not easy to obtain and not always discriminant (like pulmonary systolic pressure attained with tricuspid regurgitation jet peak velocity, left atrial enlargement, and mitral E/A ratio ). In normal subjects E/e’ is not reliable . This happens for the protodiastolic increase suction forces that abnormally increase protodiastolic gradient by decreasing LV pressure (leaving unchanged the left atrial pressure); it follows that all diastolic waves (both pulmonary and mitral) disproportionally get higher; and consequently E-A ratio gets higher and S-D ratio and systolic fraction of pulmonary venous flow get lower; E/e’ ratio also gets higher (since e’ remains constant). But the mean atrial pressure and LV preA pressure is very low. In case of an ambiguous situation like that the PVF remains the only direct way to clarify the conundrum; it consists in measuring the Arev peak and calculating A-Arev that in case of normality should be <35 cm/s and ≤ 0 respectively. We agree with the guidelines that if you combine E/e’ >10-12 (or even higher) with the normality of other suggested parameters like pulmonary systolic pressure (<35 mmHg) as obtained with tricuspid regurgitation jet and atrial dimension, we should conclude that the filling pressure is normal. But that is only an indirect assessment! The A-Arev based on pulmonary venous flow remains the only direct evaluation age-independent that if properly recorded, is much more robust.

In conclusion E/e’ is a pretty robust parameter but has got many pitfalls and should be replaced with other echocardiographic and even invasive measurements under common clinical scenario as recently suggested [8]. We think that pulmonary venous flow if properly recorded ( it, now, can be obtained with high feasibility with a TTE approach) is potentially better than E/e’ in DCM patients in order to predict LV filling pressure and functional capacity.

So we have largely implemented this point in the manuscript (in the Discussion under limitations subheadings pag 16 in red): “We did not measure systematically the E/e’. This is a pretty robust parameter [9] but has got many pitfalls and should be replaced with other echocardiographic and even invasive measurements under common clinical scenario as recently suggested [8]. In particular in DCM it does not predict an isolated telediastolic increase of pressure like Arev-A does, a common finding in our study group (Table 2); in fact E/e’ is and index of mean pulmonary wedge pressure [4]. It does not work properly in patients with hart failure[10]; also it is imprecise in left bundle branch block (LBBB), that was pretty common in our study group (the 2 cases, Figure 3 an 4, in the paper have got a LBBB) and in mitral annulus calcification that was also common in our patients. We think that pulmonary venous flow if properly recorded as it happens especially with the usage of contrast [5] in combination with trans-mitral flow, is a better method than E/e’ in order to predict LV filling pressure and functional capacity in DCM patients.”

Reviewer#7: Mitral regurgitation was assessed as jet area. Why not by PISA method which is recommended for quantitative assessment?

Response#7: Thanks for this comment. We evaluated mitral regurgitation during TEE study with a fast semiquantitative method because PISA method could have prolonged the exam duration and can be unfeasible for eccentric jets; since the TEE approach is in general no so well tolerated type of study we tried to shorten the exam duration as much as possible; a quantitative approach would have consistently prolonged the exam ; finally the evaluation of the mitral regurgitation severity on a semiquantitative assessment has been the policy of other TEE studies[11]. In details and with validated method mitral regurgitation was visually estimated on a semiquantitative scale according to the maximum length and width of the abnormal jet relative to the left atrium. The scan plane yielding the least left atrial foreshortening and the largest color flow jet was used to grade regurgitation as follows: 0 indicated no regurgitation,1+ was assigned if the regurgitant jet was less than one third of both the length and width of the left atrium; 2+ if the jet was one third to one half of the length and width of the left atrium; 3+ if the jet was one half to two thirds of the length and width of the left atrium; and 4+ if the jet exceeded two thirds of the length and width of the left atrium. These criteria were applied to central and eccentric regurgitant jets having proportional lengths and widths. When eccentric jets having length and widths disproportionate to one another were noted, the maximal severity of regurgitation based on either length or width was assigned and then reduced by one degree [11,12].

So in the method section we have synthetically added some details regarding the mitral regurgitation assessment (page   in red): “Mitral regurgitation was visually estimated on a semiquantitative scale according to the maximum length and width of the abnormal jet relative to the left atrium as visualized in 4-chamber view [11,12]. Mitral regurgitation was considered severe in case the ratio of area jet to left atrium area was >50% and at same time the jet caused a reversal of systolic flow in the pulmonary vein(s). Patients with severe regurgitation were excluded from the study.”

Reviewer#8: Why patients were divided above and below 14 ml/kg/min peak VO2? How were distributed parameters according to Weber's classification?

Response#8: Thanks for asking. A peak Vo2 value of 14 ml/Kg/min is a largely accepted cutoff to separate DCM patients with a discrete/good functional capacity (>14 ml/Kg/min) from those with a bad one (≤14 ml/Kg/min). In fact such a cutoff has got very important implication in predicting prognosis as expressed in the landmark study by Mancini et al. [13]. The Weber classification is absorbed in this dichotomic classification of max Vo2. In fact Weber classification consider a class A when max Vo2 exceeds 20/ml/Kg/min ; mild to moderate impairment , termed class B, is present when Vo2 max ranges between 16 and 20 ml/Kg/min.; a moderate to severe impairment, class C, exists whem Vo2 max fall between 10 and 16 ml/min/Kg. Class D represents a severe impairment, with Vo2 max ranging between 6 and 10 ml/min/Kg. So our dichotomic classification largely accepted in the literature includes in the good arm (max Vo2 >14 ml/Kg/min) subjects with a Weber class A and B and some of those included in class C; while in bad arm (max Vo2 ≤ 14 ml/Kg/min) part of the C group and all those belonging to the real bad one (class D). To give a better flavor of the meaning of this dicotomic classification of max Vo2 we have added in the Table 3 as a continuous variable the same Max Vo2. .

Reviewer#9: There is a lack of cardiopulmonary exercise test parameters: peak RER, VO2 at anaerobic threshold, VE/VCO2 slope, percent predicted peak VO2, Heart rate response, chronotropic index.

Response#9: We appreciate our legitimate question. However all the parameters that you have mentioned can be useful in assessing the progression, response to medical therapy, improvement in cardiovascular fitness with training and most of all to predict prognosis [14].

Contrary MaxVo2 defines functional capacity; in particular functional capacity is the ability of an individual to perform aerobic work as defined by the maximal oxygen uptake; it is the product of cardiac output and arterio-venous oxygen difference at physical exhaustion.

Minor points:

Reviewer#10: Exclusion criteria: “Exclusion criteria were the presence of acute cardiac or extracardiac illness, atrial fibrillation or any cardiac arrhythmia, severe valvulopathies. (We also excluded severe mitral valve regurgitation because this affects LV diastolic filling, causing an increased E-wave velocity).” Severe valvulopathies include severe MR. Criteria for severe mitral regurgitation should be described.

Response#10: Thanks for this question. We totally agree with reviewer; so we have added the criteria in the method section page 3 in red : “Mitral regurgitation was visually estimated on a semiquantitative scale according to the maximum length and width of the abnormal jet relative to the left atrium as visualized in 4-chamber view [11,12]. Mitral regurgitation was considered severe if the ratio of area jet to left atrium area was >50% along with reversal of systolic flow in the pulmonary vein(s). Patients with severe regurgitation were excluded from the study.”

Reviewer#11: English should be improved.

Response#11:

We tried to improve English and the paper was reviewed by a mother tongue expert

References

  1. Nagueh SF, Smiseth OA, Appleton CP, Byrd BF, 3rd, Dokainish H, Edvardsen T, et al. Recommendations for the Evaluation of Left Ventricular Diastolic Function by Echocardiography: An Update from the American Society of Echocardiography and the European Association of Cardiovascular Imaging. Eur Heart J Cardiovasc Imaging. 2016;17(12):1321-60.
  2. Rossvoll O, Hatle LK. Pulmonary venous flow velocities recorded by transthoracic Doppler ultrasound: relation to left ventricular diastolic pressures. Journal of the American College of Cardiology. 1993;21(7):1687-96.
  3. Kuecherer HF, Muhiudeen IA, Kusumoto FM, Lee E, Moulinier LE, Cahalan MK, et al. Estimation of mean left atrial pressure from transesophageal pulsed Doppler echocardiography of pulmonary venous flow. Circulation. 1990;82(4):1127-39.
  4. Nagueh SF, Appleton CP, Gillebert TC, Marino PN, Oh JK, Smiseth OA, et al. Recommendations for the evaluation of left ventricular diastolic function by echocardiography. Journal of the American Society of Echocardiography : official publication of the American Society of Echocardiography. 2009;22(2):107-33.
  5. Masuyama T, Nagano R, Nariyama K, Lee JM, Yamamoto K, Naito J, et al. Transthoracic Doppler echocardiographic measurements of pulmonary venous flow velocity patterns: comparison with transesophageal measurements. Journal of the American Society of Echocardiography : official publication of the American Society of Echocardiography. 1995;8(1):61-9.
  6. Dini FL, Michelassi C, Micheli G, Rovai D. Prognostic value of pulmonary venous flow Doppler signal in left ventricular dysfunction: contribution of the difference in duration of pulmonary venous and mitral flow at atrial contraction. Journal of the American College of Cardiology. 2000;36(4):1295-302.
  7. Lambertz H, Schuhmacher U, Tries HP, Stein T. Improvement of pulmonary venous flow Doppler signal after intravenous injection of Levovist. Journal of the American Society of Echocardiography : official publication of the American Society of Echocardiography. 1997;10(9):891-8.
  8. Park JH, Marwick TH. Use and Limitations of E/e' to Assess Left Ventricular Filling Pressure by Echocardiography. J Cardiovasc Ultrasound. 2011;19(4):169-73.
  9. Nagueh Sherif F. Left Ventricular Diastolic Function. JACC: Cardiovascular Imaging. 2020;13(1_Part_2):228-44.
  10. Mullens W, Borowski AG, Curtin RJ, Thomas JD, Tang WH. Tissue Doppler imaging in the estimation of intracardiac filling pressure in decompensated patients with advanced systolic heart failure. Circulation. 2009;119(1):62-70.
  11. Sheikh KH, Bengtson JR, Rankin JS, de Bruijn NP, Kisslo J. Intraoperative transesophageal Doppler color flow imaging used to guide patient selection and operative treatment of ischemic mitral regurgitation. Circulation. 1991;84(2):594-604.
  12. Smith MD, Harrison MR, Pinton R, Kandil H, Kwan OL, DeMaria AN. Regurgitant jet size by transesophageal compared with transthoracic Doppler color flow imaging. Circulation. 1991;83(1):79-86.
  13. Mancini DM, Eisen H, Kussmaul W, Mull R, Edmunds LH, Jr., Wilson JR. Value of peak exercise oxygen consumption for optimal timing of cardiac transplantation in ambulatory patients with heart failure. Circulation. 1991;83(3):778-86.
  14. Arena R, Myers J, Williams MA, Gulati M, Kligfield P, Balady GJ, et al. Assessment of functional capacity in clinical and research settings: a scientific statement from the American Heart Association Committee on Exercise, Rehabilitation, and Prevention of the Council on Clinical Cardiology and the Council on Cardiovascular Nursing. Circulation. 2007;116(3):329-43.

Reviewer 2 Report

I have read with great interest your manuscript PULMONARY VENOUS FLOW IN PREDICTING FUNCTIONAL CAPACITY IN DILATED CARDIOMYOPATHY. A 
TRANSESOPHAGEAL ECHOCARDIOGRAPHIC DOPPLER STUDY 

The abstract is not well structure

Material and Methods: The design shows various flaws 

The discussion section just settles your own results, there is a lack of argument and do not drive to the conclusion. 

Tables, legends and abbreviations need to be revised. Follow are the details:

PULMONARY VENOUS FLOW IN PREDICTING FUNCTIONAL CAPACITY IN DILATED
CARDIOMYOPATHY. A TRANSESOPHAGEAL ECHOCARDIOGRAPHIC DOPPLER STUDY

The design of the study is confusing. 119 (122?) patients included, 27 without clinical capacity
to perform an exercise. Afterwards, patients are divided in two groups based on the VO2 and,
these former 27 were also include in this group. All patients were performed a TEE but several
parameters could not be evaluated because of the overlapping of E and A wave.

Patients included are in sinus rhythm, but 15 pts need could not be completely assessed
because an overlapping E, A wave, meaning that one of the recorded parameter with the TEE
has been influence by this loss of patients? But did this tachycardia not influence on the rest of
the measurements? Usually it does.

What happened with all the transthoracic echocardiogram information? Do the authors
consider that perhaps atrial volume or Doppler Tissue Imaging E/e ́ratio could be of interest to
relate with the TEE data? Moreover, New York Functional Class although the manuscript says
that has been recruited, is not used. Consider that is one of the most useful tool for the clinical
cardiologist.

Patients with several comorbidities or acute heart failure should not be included in the study.
And the inclusion of the 24 patient based on the functional class, but in which group were
these patients included? It is a serious methodological flaw.

Extracardiac factors influence in all cardiovascular pathology. Their use in this manuscript is
confusing.

Obesity has been said that expand the blood volume; please explain this confusing
information.

Chronic kidney disease patients are haemodialysis? How many of them?

The authors focus on Kidney insufficiency and obesity. But what about several factors that
influence on the diastolic pattern regulation such as these influencing on preload, afterload
and factors influencing Frank Starling law?

The discussion section is a confirmation of the obtained results; there is a lack of argument.

Other suggestions:

Improve tables and Figures. Add a flowchart to understand the design of the study.

Several articles used in the article has more than 20 years.

Legends need to be revised

Abbreviations are wrongly used along the text.

Author Response

ANSWERS.

Reviewer#1: The design of the study is confusing. 119 (122?) patients included, 27 without clinical capacity
to perform an exercise. Afterwards, patients are divided in two groups based on the VO2 and,
these former 27 were also include in this group. All patients were performed a TEE but several
parameters could not be evaluated because of the overlapping of E and A wave.
Patients included are in sinus rhythm, but 15 pts need could not be completely assessed
because an overlapping E, A wave, meaning that one of the recorded parameter with the TEE
has been influence by this loss of patients? But did this tachycardia not influence on the rest of
the measurements? Usually it does.
.

Response#1: Very good question, thanks for it. Our lost of cases regarding E/A ratio parameter for the fusion problem was not so crucial: we loose just a little power in the statistics. The good point is if the tachycardia in these 15 patients of ours with E/A fusion can affect pulmonary venous flow waves. In one study with E -A fusion and tachycardia (patients in intensive care unit) pulmonary venous flow and in particular SF (systolic fraction) were the second best parameters in ‘predicting pulmonary capillary wedge pressure (PCWP): in patients with satisfactory pulmonary venous flow Doppler recordings, SF had the best correlation with PCWP (r =0.53, P =0.009) that was the second best correlation after E/e’. In this study the best pulmonary parameter to predict a PCPW >15 mmHg was a SF < 40% with a sensitivity and specificity of 67 and 88% respectively [1]. However in Nagueh’s study the quality of PVF Doppler was suboptimal while in our study was very good. So we would expect that SF would have worked even better in our study in the 15 patients with sinus tachycardia and E-A fusion in predicting PCWP and consequently functional capacity.

Reviewer#2: Patients with several comorbidities or acute heart failure should not be included in the study. And the inclusion of the 24 patient based on the functional class, but in which group were these patients included? It is a serious methodological flaw.

Response#2: Thanks for this appropriate observations. When the patients were recovered in our rehabilitation center they were all in adequate conditions to perform both CPX evaluation and the TEE study ; but being a fragile group of patients their clinical condition unfortunately in some cases got worse during the recovery so they were unable to perform the CPX test that in general was scheduled after the TEE study. The main reasons for that were arrhythmias, angina, pulmonary and other infections, in one case pneumothorax, CPOD worsening, acute heart failure in a few cases, psychiatric disturbances. In case the CPX was not done for the reasons explained above we carefully collected an history regarding their functional capacity and tried to classify the patients in a NYHA functional class. So both the TEE and the functional capacity assessment (either by CPX or NYHA) reflects a picture of a relatively stable conditions.

The other major problem here is if the NYHA class could be a weak surrogate of the objectively assessed max Vo2 consumption obtained by CPX. In a study that tried to compare NYHA and max V02 by CPX the authors found a certain correlations but that was not optimal [2]; in fact a lower NYHA class (II, III) tended to underestimate max Vo2 consumption that remains superior to 20 ml/Kg/min in most patients in these 2 NYHA classes. However only the minor part of our 24 patients that were not able to exercise (5 patients, 20% ) were included in the subgroup with max Vo2 < 14 ml/Kg/min on the basis of the NYHA class <3. So on the basis of the previous cited study the error that we’d eventually introduce with this NYHA classification in predicting max Vo2 should be modest.

Reviewer#3: Extracardiac factors influence in all cardiovascular pathology. Their use in this manuscript is confusing.

Response#3: Thanks again for this important comment. We agree that cardiovascular function is influenced by several extracardiac factors. Unfortunately this is not a well covered topic in the literature. In our study we focused on those common factors that we have to deal with in our clinical practice in patients with DCM. We basically focused on two extracardiac factors: obesity and renal dysfunction. These factors are important since they have been seen to worsen functional capacity but at same time we can act upon them so potentially improving cardiac-vascular performance, the clinical status and eventually the clinical outcome of our patients. We agree that other extracardiac factors can act on cardiac performance (high output states like tyrotossicosis etc infection, sepsis , pulmonary disorders etc) but that was not the case of our chronic DCM patients in stable conditions.

Reviewer#4: Obesity has been said that expand the blood volume; please explain this confusing information.

Response#4: Thanks for asking for this pathophysiologic insight. Considerable evidence demonstrates the adverse effects of obesity on central and peripheral hemodynamics, as well as on cardiac structure and function. In particular with exercise in class III obese patients, central blood volume increases by 20%, LV end-diastolic pressure increases by 50% (from an already elevated 21 to 31 mm Hg), and LV dP/dt increases 57% [3]. Fatfree (non-osseous) mass is thought to contribute to these alterations as augmentation of total blood volume, and cardiac output cannot be accounted for by excess fat mass alone [3]. This happens in severe obesity but without a dilated cardiomyopathy. So it is easy to imagine that in patients with already intrinsic cardiac dysfunction as in patients with DCM, even less severe obesity like that affected our study group, through a milder increase of central blood volume, creates higher LV filling pressure so reducing filling and cardiac output in this patients during effort so at the end reducing maximal functional capacity.

So we have implemented the manuscript in the discussion (page 16 in red):” Considerable evidence demonstrates the adverse effects of obesity on central and peripheral hemodynamics, as well as on cardiac structure and function. In particular with exercise in class III obese patients, central blood volume increases by 20%, LV end-diastolic pressure increases by 50% (from an already elevated 21 to 31 mm Hg), and LV dP/dt increases 57%. Fatfree (non-osseous) mass is thought to contribute to these alterations as augmentation of total blood volume, and cardiac output cannot be accounted for by excess fat mass alone. Thus….”

Reviewer#5: Chronic kidney disease patients are haemodialysis? How many of them?

Response#5: Thanks for this question. None of our patients were on hemodialysis. This important peace of information along with renal insufficiency data was added in the Table 1.

Reviewer#6: The authors focus on Kidney insufficiency and obesity. But what about several factors that influence on the diastolic pattern regulation such as these influencing on preload, afterload
and factors influencing Frank Starling law?

Response#6: Thanks again for stressing again the role of extracardiac factors. We think that we have tried to cover this point in the anwer #3.

Reviewer#7: The discussion section is a confirmation of the obtained results; there is a lack of argument.

Response#7: We really appreciate this comments. You are right, we expanded that part of the discussion related to the explantions of our findings. The eventually contrasting findings were really a few since PVF has never been tried with TEE approach in particular to predict functional capacity; in addition extracardiac factor has never been covered in depth in this context. However a synthetic covering of other reports on the topic of functional capacity predictors have been reported in the “previous studies” section of the discussion.

Reviewer#8: Other suggestions: Improve tables and Figures.

Response#8: Thanks for this useful suggestion; I reviewed all the tables and Figures and I made some legends corrections. For best readability I also added some subheadings (Tables 1 and 2) .

Reviewer#9. Add a flowchart to understand the design of the study.

Response#9: Thanks for this suggestion; we agree with you so we have added a flow chart in the manuscript (Figure 2)

Reviewer#10: Several articles used in the article has more than 20 years. Legends need to be revised ; Abbreviations are wrongly used along the text.

Response#10: Thanks for this final comment. We have added more recent articles and we have checked all the legends and abbreviations.

  1. Nagueh SF, Mikati I, Kopelen HA, Middleton KJ, Quinones MA, Zoghbi WA. Doppler estimation of left ventricular filling pressure in sinus tachycardia. A new application of tissue doppler imaging. Circulation. 1998;98(16):1644-50.
  2. Rostagno C, Galanti G, Comeglio M, Boddi V, Olivo G, Gastone Neri Serneri G. Comparison of different methods of functional evaluation in patients with chronic heart failure. European journal of heart failure : journal of the Working Group on Heart Failure of the European Society of Cardiology. 2000;2(3):273-80.
  3. Alexander JK. Obesity and cardiac performance. The American journal of cardiology. 1964;14(6):860-5.

Round 2

Reviewer 1 Report

I accept in present form.

Reviewer 2 Report

The design of the study is not correct 

Although the manuscript has been changed, the most important flaws such as the design that leads to the results are not possible to discuss.